# The Relation between CEO-Friendly Boards and the Value of Cash Holdings

**Hoontaek Seo [1,\*], Sangho Yi [2], Qing Yang [1] and William McCumber [3]**

[1]   Holzschuh College of Business Administration, Niagara University, NY 14109, USA
[2]   Sogang Business School, Sogang University, Seoul 04107, Republic of Korea
[3]   College of Business, Lousiana Tech University, Ruston, LA 71272, USA
[\*]   Correspondence: hseo@niagara.edu

**Abstract:** Our study investigates how CEO-friendly boards influence the value and utilization of cash resources. In this paper, we analyze two conflicting views on CEO-friendly boards and their impact on corporate cash holdings: one view posits that such boards might be too lenient, fostering managerial moral hazard problem, while the other contends that they encourage CEOs to share information, despite CEOs knowing that better-informed boards could enforce stricter oversight. By measuring board friendliness through CEO-board social ties, we find that firms with a friendly board tend to maintain lower cash reserves but their excess cash is valued higher by the market compared to firms without such a board. Moreover, these boards deploy excess cash in ways that significantly enhance firm value. The results remain robust even after controlling for various governance variables and CEO characteristics. Our findings offer crucial insights for corporate practitioners and policymakers, highlighting the importance of appointing and retaining CEO-friendly directors to foster effective information exchange, especially in firms with substantial CEO-board information asymmetry in capital budgeting.

**Keywords:** friendly boards; cash holdings; excess cash; investment efficiency

**JEL Classification:** G30

## 1. Introduction

Firms are known to maintain substantial cash reserves, a practice that may appear puzzling considering the typically lower returns on cash compared to those from other investments (Bates et al. 2009; Duchin 2010). The decision to hold cash, like other corporate choices, is guided by a marginal benefit–marginal cost analysis (Saunders et al. 2021). However, some previous studies suggest that a firm's cash holding decisions may deviate from the optimal level suggested by the marginal benefit–marginal cost approach (Jensen 1986; Lang et al. 1991). Given the importance of liquid asset management in corporate finance, it is unsurprising that numerous studies have investigated the determinants of corporate cash holdings, aiming to identify factors that could significantly influence a firm's motivation to maintain a certain amount of cash, as well as the value implications of the size of corporate cash reserves (Opler et al. 1999; Han and Qiu 2007; Acharya et al. 2012; Harford et al. 2014; Marwick et al. 2020).

The managerial agency problem is one such factor (Jensen 1986; Faulkender and Wang 2006). Principal-agent theory posits that, in the presence of information asymmetry and incomplete contracts, managers often have an incentive to prioritize their private benefits Principal–agent theory suggests that CEOs, facing significant interest and information gaps with directors, may prioritize personal benefits over shareholder interests, sometimes to the detriment of shareholders (Jensen and Meckling 1976; Tirole 2006)[1]. These private benefits of control for CEOs can manifest in various ways[2]. Many of these activities by managers,

aimed at enhancing the CEO's private control-related benefits, become more likely when a firm has a substantial pool of cash.

Jensen (1986) develops this idea and proposes the agency costs of free cash flow hypothesis. This hypothesis suggests that when managers possess excess cash beyond what is required for funding positive NPV projects (i.e., free cash flow), they may have an incentive to squander it on unprofitable investments like acquisitions which increases their private benefits (Lang et al. 1991). Furthermore, ample free cash flow may reduce the pressure on management to achieve specific performance goals, thereby also exacerbating the managerial moral hazard problem[3]. Several prior studies have supported the free cash flow hypothesis by reporting empirical findings that confirm a negative relationship between the size of free cash flow, when interacting with CEO moral hazard incentives, and firm value (Lang et al. 1996; Lang et al. 1991). Among various corporate governance mechanisms, outside directors, who are independent of CEO influence and possess a stronger incentive to monitor and, if necessary, discipline the CEO, tend to enhance the value contributions of corporate cash (Hsu et al. 2015). This serves as further evidence that effective corporate governance mechanisms mitigate the cost of managerial moral hazard, particularly when exacerbated by the misallocation of free cash flows.

In this paper, we adopt a comprehensive approach to explore how directors influence the value generated from corporate cash while continuously interacting with a self-interested CEO. This is important since, as Adams et al. (2010) suggest, directors need information revealed by the CEO to provide valuable services to a firm including advising, participating in business strategy formation, CEO hiring/firing/assessment, and taking action during the interim period when a CEO replacement is imminent. The amount of information revealed by a CEO depends on the CEO's voluntary decision. Here, the CEO faces trade-offs. If the CEO reveals more information, it can enhance advisory services that ultimately benefit the CEO. However, the information disclosed also empowers directors to monitor the CEO more effectively. Thus, the provision of valuable services by directors to a firm can be likened to a continuous bargaining game between the CEO and the directors.

As Adams and Ferreira (2007) suggest, CEO-friendly boards can provide an optimal solution for this game. Harris and Raviv (2008), Raheja (2005), and Schmidt (2015) argue that shareholders prefer CEO-friendly boards when the advisory roles of the board of directors are more important than their monitoring roles. On the other hand, concerns arise about exacerbating the managerial moral hazard problem since CEO-friendly boards, who often have social ties with the CEO, are typically less inclined to engage in aggressive monitoring and disciplining actions (Adams and Ferreira 2007). Given that previous studies have provided support for both directions in the relationship between CEO-friendly boards and CEO's moral hazard problem, the impact of CEO-friendly boards on the value contributions of corporate cash is ultimately an empirical matter.

Previous studies indicate that when CEOs and directors have strong ties, outside directors may become overly familiar with the CEO, possibly leading to reluctance to question their decisions or uncover any misconduct. These connections can undermine the effectiveness of the board and exacerbate CEO opportunism, resulting in suboptimal investment behaviors and distorting corporate investment decisions (Cohen et al. 2008; Fracassi and Tate 2012; Duchin and Sosyura 2013). Conversely, strong ties can benefit shareholders by facilitating valuable information exchange between the CEO and the board. Research indicates that strong CEO–director ties promote trust and transparency, fostering information sharing among them, thereby effective decision making and oversight (Cao et al. 2015; Adams and Ferreira 2007; Cai et al. 2015).

Another line of research presents contradictory findings regarding the relationship between corporate governance, corporate cash holding practices, and the value of cash to shareholders. For instance, Dittmar and Mahrt-Smith (2007) demonstrate that in poorly governed firms, the value of excess cash diminishes, and these firms tend to allocate excess cash towards less profitable investments compared to firms with strong corporate governance. Pinkowitz et al. (2006), along with Kalcheva and Lins (2007), offer similar

insights from their cross-country analyses. In contrast, Opler et al. (1999) find no significant association between corporate cash holding practices and firm-level corporate governance. Thus, the causal relationship between corporate governance and corporate cash holding practices remains unclear in existing literature, representing an ongoing empirical inquiry.

In this paper, we propose two competing hypotheses concerning whether CEO-friendly boards enhance the value contributions of corporate cash. The first one is what we call the "aggravated free cash flow" hypothesis. This hypothesis suggests that the presence of CEO-friendly boards who are favorable and lenient toward the CEO only amplifies the CEO's incentive to misuse the firm's free cash flows for personal gain. The second hypothesis is what we call the "efficient advisory" hypothesis. Since CEO-friendly boards can foster effective communication with the CEO and encourage the voluntary disclosure of information, this can enhance the quality of the directors' advisory services. As these advisory services become more excellent and reliable, the value of corporate cash may increase accordingly because the CEO can more effectively identify projects with higher NPV that are difficult to notice. It is also possible that better-informed CEO-friendly boards can more adeptly identify projects with negative NPV that enhance the CEO's personal benefits. Information Economics also confirms this prediction (Holmström 1979; Mirrlees 1999). Holmström (1979) suggests that the creation of additional information systems that facilitate the production of information, such as a CEO-friendly board, can improve the efficiency of a CEO's decisions in capital budgeting. This improvement will directly translate to an enhanced firm.

In this paper, we test these two competing hypotheses to deepen our understanding of how CEO-friendly boards influence the CEO's incentive to allocate cash across various projects. Additionally, this relates to another objective of understanding the board of directors from a broader and more comprehensive perspective, as opposed to a narrow focus on inside/outside directorship. Adams et al. (2010) point out that the prevailing focus of extant studies on inside/outside directorship is primarily motivated by a statistical reason: to address the endogeneity problem that is so prevalent in studies on the board of directors. Recently, a growing number of studies have focused on various aspects of the board of directors. The CEO-friendliness of directorship is an important topic investigated by them.

Our empirical results support the efficient advisory hypothesis. First, we find that firms with a CEO-friendly board tend to hold less cash. This is consistent with the efficient advisory hypothesis. As the board's effectiveness in providing advice improves, the efficiency of cash utilization increases, and the opportunity costs of corporate cash holdings rise for firms with a CEO-friendly board. Consequently, this leads to a reduction in cash holdings for firms with a CEO-friendly board. Second, we find that the market value of excess cash reserves is greater for firms with a friendly board. Third, we find that the utilization of excess cash has a negative impact on firm value for firms without a CEO-friendly board. In contrast, firms with a CEO-friendly board demonstrate a more profitable use of excess cash. Finally, we find that investment sensitivity to Q is higher in firms with a friendly board compared to firms without a friendly board. This suggests that friendly boards can enhance investment efficiency by facilitating directors to offer advice, thereby improving management's ability to select better projects. Overall, our results align with the efficient advisor hypothesis.

Our study makes three distinct contributions to the literature on cash holdings and CEO-friendly boards. Firstly, we introduce and provide empirical support for the concept of CEO-friendly directorship as a novel determinant of cash holdings, a perspective previously unexplored in the literature. While prior research has identified various factors influencing corporate cash holdings, such as firm size (Opler et al. 1999), leverage (Ferreira and Vilela 2004), corporate governance (Dittmar and Mahrt-Smith 2007), corporate social responsibility (Cheung 2016), and CEO overconfidence (Chen et al. 2020), the influence of CEO-friendly directorship on cash holdings has not been investigated.

Secondly, we explore the impact of a crucial characteristic of the board of directors, beyond inside–outside directorship, on the value of corporate cash holdings and the efficiency of CEOs in allocating cash to various projects. In doing so, we illuminate a novel mechanism by which corporate governance arrangements contribute to an increase in firm value. Efficient information flow from the CEO to CEO-friendly boards enhances the quality of directors' advisory services. Considering the interaction between directors offering advice to the CEO and the voluntary disclosure of information by the CEO as a bargaining game, our study posits that minimizing the bargaining costs in this dynamic will ultimately increase the value of corporate cash holdings.

Finally, we contribute to the rapidly expanding literature on the economic impacts of CEO-friendly boards, as we introduce a significant and previously unexplored effect of CEO-friendly boards on corporate cash holdings[4].

The rest of the paper is organized as follows: Section 2 develops the main hypotheses. Section 3 describes the data and sample. Section 4 presents the empirical results. Section 5 concludes the paper.

## 2. Hypothesis Development

In this section, we develop hypotheses that will be empirically examined in this paper. One hypothesis to be tested is termed the "aggravated free cash flow" hypothesis. This hypothesis proposes that lenient and CEO-friendly boards are hesitant to take aggressive actions against the CEO, incentivizing the CEO to hold more cash that he may misuse to pursue his private benefits of control. The misappropriation of cash by CEOs for private gain will reduce firm value through two channels. First, due to the pursuit of private benefits by CEOs, the company may invest in projects with low NPV or even negative NPV, as long as those projects increase the CEO's private benefits. Second, holding more cash will increase the opportunity cost of cash holdings due to the low return on cash.

The second hypothesis, referred to as the "efficient advisory" hypothesis, posits that CEO-friendly boards enhance the CEO's information disclosure, thereby improving the quality of directors' advisory services and, consequently, the value contributions of corporate cash as well. The efficient advisory hypothesis suggests that in firms with a friendly board, the efficiency of cash utilization will increase as the board provides more effective advice, leading to a reduction in cash holdings. Given the high opportunity costs associated with cash holdings, a decrease in cash holdings will result in an increase in firm value. Therefore, we suggest the following hypothesis:

**H1.** *The efficient advisory hypothesis posits a negative relation between CEO-friendly boards and the level of cash holdings. In contrast, the aggravated free cash flow hypothesis anticipates a positive relation between CEO-friendly boards and the level of cash holdings.*

Our next hypothesis is about the impact of CEO-friendly boards on the value of cash holdings. The efficient advisory hypothesis suggests that CEO-friendly boards enhance the efficiency of the CEO's cash allocation across various projects, as these directors facilitate information flow from the CEO[5]. Conversely, the aggravated free cash flow hypothesis posits that the presence of CEO-friendly boards diminishes the value contributions of corporate cash due to their less intensive monitoring of the CEO. Therefore, we propose the following hypothesis:

**H2.** *The efficient advisory hypothesis predicts that CEO-friendly boards have a positive impact on the value of excess cash holdings. In contrast, the aggravated free cash flow hypothesis anticipates that CEO-friendly boards have a negative impact on the value of excess cash holdings.*

Building on the previous hypotheses, we can further develop the following hypothesis regarding how CEO-friendly boards impact the profitability of the use of excess cash:

**H3.** *The efficient advisory hypothesis predicts that CEO-friendly boards have a positive effect on the profitability of the excess cash utilization. In contrast, the aggravated free cash flow hypothesis anticipates that CEO-friendly boards have a negative effect on the profitability of excess cash utilization.*

Our final hypothesis relates to the impact of CEO-friendly boards on investment efficiency. The efficient advisory hypothesis suggests that CEO-friendly boards assist the CEO in making investment decisions more efficiently, thereby enhancing firm value. Conversely, the aggravated free cash flow hypothesis posits that CEO-friendly boards contribute to a decline in the efficiency of the CEO's investment decisions, as they exacerbate the CEO's moral hazard problem. Therefore, we develop the following hypothesis:

**H4.** *The efficient advisory hypothesis predicts that CEO-friendly boards enhance investment efficiency. In contrast, the aggravated free cash flow hypothesis anticipates that CEO-friendly boards worsen investment efficiency.*

### 3. Data Description

*3.1. Sample Construction*

Our initial sample includes all firms covered in the BoardEx database from 1999 to 2019. The BoardEx database contains information on board structures and detailed profiles of individual executives and board members. The available information for each director in the BoardEx database includes current and previous employment details, educational background, affiliations with not-for-profit associations, and club memberships. We remove firms that are utilities (SIC 4900-4999) or financials (SIC 6000-6999). We obtain accounting data from Compustat, stock return data from CRSP, executive ownership and compensation data from ExecuComp, and institutional holdings from the Thomson-Reuters 13f filing Database. Our final sample consists of 21,471 firm-year observations.

*3.2. Measure of Board Friendliness*

Motivated by Schmidt (2015) and Kang et al. (2018), a board is classified as a friendly board if at least one of the outside directors is socially connected to the CEO. We consider the CEO and a board member as socially connected if they attended the same school and graduated within two years of each other. Additionally, we classify two individuals as socially connected if they share a present or past membership in the same charity, club, or other non-profit association (Bruynseels and Cardinaels 2014). To represent these social connections, we employ a binary indicator variable that takes a value of one when the board is considered friendly and zero otherwise.

*3.3. Descriptive Statistics*

All variables in dollars are inflation-adjusted to 2019 dollars using the consumer price index (CPI). All continuous variables are winsorized at the 1st and 99th percentiles in order to minimize the impact of outliers. We define all of our variables in Appendix A. Table 1 presents the mean and median values of the key variables in this study. The comparison of firm characteristics shows that firms with a friendly board are larger and have lower cash-to-sales ratios than firms without a friendly board. Interestingly, firms with a friendly board have a significantly lower market-to-book ratio. However, since we have not controlled for other market-to-book determinants, we will postpone making inferences until the regression analysis. We find that there is no difference in excess cash holdings between the two groups of firms. Firms with a friendly board have lower institutional ownership and managerial ownership compared to firms without a friendly board.

**Table 1.** Descriptive statistics.

| | Firms with Friendly Board (N = 7,774) | | Firms with No Friendly Board (N = 13,697) | |
|---|---|---|---|---|
| | **Mean** | **Median** | **Mean** | **Median** |
| *Firm characteristic:* | | | | |
| Cash/Sales | 0.239 | 0.095 | 0.331 *** | 0.139 *** |
| Assets | 12.741 | 3.093 | 3.695 *** | 1.046 *** |
| PP&E | 0.311 | 0.238 | 0.275 *** | 0.202 *** |
| Excess cash | 0.000 | 0.000 | 0.000 | 0.000 |
| Market-to-book | 2.562 | 1.803 | 2.961 *** | 1.858 *** |
| Capital expenditures | 0.0586 | 0.041 | 0.058 | 0.039 *** |
| Investment | 0.031 | 0.006 | 0.044 *** | 0.010 *** |
| Q | 2.099 | 1.702 | 2.199 *** | 1.710 ** |
| CF | 0.138 | 0.124 | 0.137 | 0.127 *** |
| Institutional ownership | 22.237 | 20.935 | 24.586 *** | 24.586 *** |
| Insider ownership | 2.598 | 0.360 | 3.465 *** | 0.731 *** |
| *Board and CEO characteristics:* | | | | |
| Board size | 12.345 | 13.000 | 10.126 *** | 10.000 *** |
| CEO duality | 0.635 | | 0.424 *** | |
| CEO age | 56.411 | 56.000 | 55.537 *** | 55.000 *** |
| CEO gender | 0.956 | | 0.974 *** | |
| CEO tenure | 8.668 | 6.252 | 7.842 *** | 5.586 *** |
| CEO wealth sensitivity | 525.289 | 6.515 | 84.375 ** | 5.430 *** |

This table reports the summary statistics for the key variables used in our analysis. Variable definitions are in the Appendix A. The significance of the mean (median) difference is assessed using a t-test (Wilcoxon test). The labels *, **, and *** indicate statistical significance at the 1%, 5%, and 10% levels, respectively.

The comparison of board characteristics shows that directors of firms with a friendly board are older, more likely to be female, and have longer tenure compared to directors without a friendly board. We find that firms with a friendly board have significantly larger boards. We also find that the CEO is more likely to be the board chairman in a firm with a friendly board. CEO wealth-performance sensitivity in firms with a friendly board is significantly higher than in firms without a friendly board.

## 4. Empirical Results

### 4.1. Friendly Boards and Cash Holdings

In this section, we investigate the relationship between friendly boards and corporate cash holdings in a multivariate setting. Our aim is to determine if cash levels at firms with a friendly board differ significantly from those at firms without a friendly board. To explore this relationship, we adapt the cash holding models of Opler et al. (1999) and Chen et al. (2020) and include the friendly board indicator as an independent variable in our model. The regression equation is specified as follows:

$$
\begin{aligned}
Cash\ holdings_{i,t} = \ & \beta_0 + \beta_1\ Friendly\ board_{i,t} + \beta_2\ Assets_{i,t} + \beta_3 NWC_{i,t} + \beta_4 Cash\ Flow_{i,t} \\
& + \beta_5 Cash\ Flow\ Volatility_{i,t} + \beta_6 Market\ to\ book_{i,t} \\
& + \beta_7 Capital\ Expenditures_{i,t} + \beta_8 Acquisition\ Expenditures_{i,t} \\
& + \beta_9 Leverage_{i,t} + \beta_{10} RD_{i,t} + \beta_{11} Dividend_{i,t} \\
& + \beta_{12} Institutional\ Ownership_{i,t} + \beta_{13} Insider\ Ownership_{i,t} \\
& + \beta_{14} CEO\ wealth\ sensitivity_{i,t} + Year\ Fixed\ Effects \\
& + Firm\ Fixed\ Effects + \varepsilon_{i,t}
\end{aligned}
\tag{1}
$$

The dependent variable is cash holdings, as measured by the natural log of cash and equivalents to sales. The right-hand-side variables are as follows: friendly board, as measured by a dummy variable equals to one if at least one of the outside directors is socially connected to the CEO and zero otherwise; real assets, as measured by total assets minus cash and equivalents deflated to 2019 dollars using the consumer price index; net working capital, as measured by current assets minus cash minus current liabilities, divided

by total assets net of cash; cash flow, as measured by earnings after interest, dividends and taxes but before depreciation, divided by total assets net of cash; cash flow volatility, as measured by the mean of the standard deviations of cash flow over ten years and a minimum of five years for firms in the same industry, as defined by two-digit SIC codes; market-to-book, as measured by the book value of total assets minus cash minus the book value of equity plus the market value of equity, divided by the book value of total assets net of cash; capital expenditures, as measured by capital expenditures scaled by total assets net of cash; acquisitions expenditures, as measured by acquisition expenditures scaled by total assets net of cash; leverage, as measured by total debt scaled by total assets net of cash; RD, as measured by the ratio of research and development expenditures to sales and zero when this value is missing; dividend, as measured by a dummy variable equal to one if the firm pays a common dividend and zero otherwise; institutional ownership, as measured by the aggregate percentage ownership of shares held by institutional investors which owns at least 5% of outstanding shares; insider ownership, as measured by the aggregate percentage ownership of common stocks held by top five executives; and CEO wealth sensitivity, as measured by dollar change in CEO wealth for a 100 percentage point change in firm value, divided by annual flow compensation[6]. We also include year and firm fixed effects in the regression.

The result of the cash levels regression estimation is presented in model (1) of Table 2. The coefficient on a *Friendly board* is negative and statistically significant at a 1% level, indicating that firms with a friendly board hold less cash. This result suggests that the natural logarithm of the cash-to-sales ratio is 0.047 lower for firms with a friendly board compared to firms without a friendly board. Given that the average of the natural logarithm of the cash-to-sales ratio is −2.263, the difference of 0.047 is economically significant. Turning to the other control variables, we find that firms with a higher market-to-book ratio hold more cash, which is consistent with Chen et al. (2020). This implies that firms with better investment opportunities would reserve more cash for investment. We also find that NWC and acquisition expenditures have negative effects on the level of cash holdings. In model (2), we also include board and CEO characteristics variables as control variables. These variables are board size, CEO duality, CEO age, CEO gender, and CEO tenure. The estimated coefficient on the *Friendly board* remains negative and significant. The results are consistent with our efficient advisory hypothesis (H1).

**Table 2.** Predicting the level of cash.

| Dependent Variable: Cash | (1) | (2) |
|---|---|---|
| Friendly board | −0.047 ** | −0.058 *** |
| | (−2.14) | (−2.57) |
| Assets | −0.006 *** | −0.006 *** |
| | (−4.96) | (−4.27) |
| NWC | −0.596 *** | −0.588 *** |
| | (−8.04) | (−7.80) |
| Cash flow | 0.209 *** | 0.243 *** |
| | (2.93) | (3.32) |
| Cash flow volatility | 0.002 | 0.002 |
| | (1.03) | (1.12) |
| Market-to-book | 0.066 *** | 0.063 *** |
| | (14.89) | (13.94) |
| Capital expenditures | −0.397 * | −0.479 ** |
| | (−1.84) | (−2.14) |
| Acquisitions expenditures | −0.622 *** | −0.604 *** |
| | (−6.86) | (−6.53) |
| Leverage | 0.224 *** | 0.243 *** |

**Table 2.** *Cont.*

| Dependent Variable: Cash | (1) | (2) |
|---|---|---|
| | (4.74) | (5.00) |
| RD | 1.495 *** | 1.570 *** |
| | (9.98) | (10.09) |
| Dividend | −0.049 * | −0.068 ** |
| | (−1.79) | (−2.42) |
| Institutional ownership | 0.001 | 0.001 |
| | (1.17) | (1.02) |
| Insider ownership | 0.001 | 0.002 |
| | (0.74) | (0.95) |
| CEO wealth sensitivity | 0.000 | 0.000 |
| | (0.65) | (0.61) |
| Board size | | −0.012 *** |
| | | (−2.77) |
| CEO duality | | −0.009 |
| | | (−0.41) |
| CEO age | | −0.002 |
| | | (−1.13) |
| CEO gender | | −0.100 * |
| | | (−1.68) |
| CEO tenure | | 0.000 |
| | | (−0.06) |
| Year and firm fixed effects | Yes | Yes |
| N | 14,320 | 13,542 |
| Adjusted $R^2$ | 77.42% | 77.79% |

This table reports the results of regressions of cash holdings on friendly boards. Variable definitions are in the Appendix A. The t-statics are reported in parentheses below each estimate. The labels *, **, and *** indicate statistical significance at the 1%, 5%, and 10% levels, respectively.

### 4.2. The Value Impact of Friendly Boards on Cash Holdings

In this section, we investigate the impact of friendly boards on the value of cash. To assess whether friendly boards influence the market valuation of cash holdings, we build upon the framework established by Dittmar and Mahrt-Smith (2007). We run a regression of market-to-book on excess cash, friendly board indicator, and an interaction between the two. To measure excess cash, a regression analysis is conducted, wherein *Cash holdings* are regressed on *Assets*, *NWC*, *Cash flow*, *Cash flow volatility*, *Market-to-book*, *Capital expenditures*, *Acquisition expenditures*, *Leverage*, *RD*, *Dividend*, *Year fixed effects*, and *Firm fixed effects*. The resulting value is then divided by sales to calculate the excess cash. The regression equation is specified as follows:

$$
\begin{aligned}
Market\ to\ book_{i,t} = \ & \beta_0 + \beta_1\ Excess\ cash_{i,t} + \beta_2\ Excess\ cash_{i,t} \times Friendly\ board_{i,t} \\
& + \beta_3 Friendly\ board_{i,t} + \beta_4 Assets_{i,t} + \beta_5\ PP\&E_{i,t} + \beta_6\ Cash\ flow_{i,t} \\
& + \beta_7 Institutional\ ownership_{i,t} + \beta_8\ Insider\ ownership_{i,t} \\
& + \beta_9 CEO\ wealth\ sensitivity_{i,t} + Year\ fixed\ effects \\
& + Firm\ fixed\ effects + \varepsilon_{i,t}
\end{aligned}
$$

(2)

To explore whether a board's friendliness is associated with a more substantial positive effect of excess cash on firm value, we interact the friendly board dummy variable with excess cash, thereby assessing the additional impact on value. The interaction term is the focus of our analysis. A positive (negative) coefficient on the interaction term between excess cash and friendly board indicates that friendly boards have a positive (negative)

impact on the value of cash holdings. A positive coefficient on the interaction variable would indicate the incremental impact on value due to the advisory role of a friendly board.

The result from the analysis of Equation (2) is presented in model (1) of Table 3. The result shows the impact of friendly boards on the value of excess cash. The estimated coefficient on the interaction term between excess cash and friendly boards is positive and statistically significant. The result indicates that the value of cash holdings is greater if the firm has a friendly board. To interpret this coefficient, let us consider a firm with one dollar of excess cash: if a friendly board had no impact on the value of this dollar, the coefficient on the interaction would be zero. However, our results demonstrate that the value of this dollar significantly increases, both statistically and economically, when the firm has a friendly board. Comparing the coefficients on excess cash alone and on the interaction, we observe that transitioning from a firm without a friendly board to a firm with a friendly board amplifies the marginal impact of excess cash on firm value by more than 35 times. In model (2), we add board and CEO characteristics variables including board size, CEO duality, CEO age, CEO gender, and CEO tenure. The coefficient on the interaction term remains positive and significant. The result confirms that friendly boards have a positive impact on the value of cash holdings. The results are consistent with our efficient advisory hypothesis (H2).

**Table 3.** Effect of friendly boards on value of excess cash.

| Dependent Variable: Market-to-Book | (1) | (2) |
|---|---|---|
| Excess cash | −57.443 *** (−6.95) | −66.265 *** (−7.73) |
| Excess cash x Friendly board | 35.115 ** (1.94) | 48.346 *** (2.59) |
| Friendly board | −0.048 (−1.08) | −0.057 (−1.22) |
| Assets | −0.038 *** (−14.51) | −0.039 *** (−14.17) |
| PP&E | 2.407 *** (11.40) | 2.302 *** (10.65) |
| Cash flow | 1.588 *** (12.69) | 1.757 *** (13.65) |
| Institutional ownership | −0.012 *** (−9.26) | −0.011 *** (−8.86) |
| Insider ownership | −0.007 ** (−2.00) | −0.006 (−1.58) |
| CEO wealth sensitivity | 0.000 ** (−2.35) | 0.000 ** (−2.43) |
| Board size | | −0.022 ** (−2.40) |
| CEO duality | | 0.062 (1.37) |
| CEO age | | −0.011 *** (−3.03) |
| CEO gender | | −0.150 (−1.22) |
| CEO tenure | | 0.008 ** (2.01) |
| Year and firm fixed effects | Yes | Yes |
| N | 14,320 | 13,542 |
| Adjusted R$^2$ | 64.92% | 65.58% |

This table reports the results of regressions of the market value on excess cash holdings and friendly boards. Variable definitions are in the Appendix. The t-statics are reported in parentheses below each estimate. The labels *, **, and *** indicate statistical significance at the 1%, 5%, and 10% levels, respectively.

*4.3. The Impact of Friendly Boards on the Use of Excess Cash*

In this section, we further examine the influence of friendly boards on excess cash by investigating whether firms with a friendly board utilize their excess cash more profitably compared to firms without a friendly board. To examine this, we run a regression of market-to-book on lagged excess cash, lagged friendly board indicator, and an interaction between the two. The regression equation is specified as follows:

$$
\begin{aligned}
Market\ to\ book_{i,t} = {} & \beta_0 + \beta_1 Lagged\ excess\ cash_{i,t} + \beta_2 Lagged\ friendly\ board_{i,t} \\
& + \beta_3 Lagged\ excess\ cash_{i,t} \times Lagged\ friendly\ board_{i,t} \\
& + \beta_4\ Lagged\ market\ to\ book_{i,t} + \beta_5 Assets_{i,t} + \beta_6\ PP\&E_{i,t} \\
& + \beta_7 Institutional\ ownership_{i,t} + \beta_8 Insider\ ownership_{i,t} \\
& + \beta_9 CEO\ wealth\ sensitivity_{i,t} + Year\ fixed\ effects \\
& + Firm\ fixed\ effects + \varepsilon_{i,t}
\end{aligned}
\tag{3}
$$

The dependent variable is market-to-book as a measure of firm value. We estimate our regression (3) for the sub-sample of firms that had positive excess cash at time t-1 and used up some of it by time t. In other words, we analyze firms that dissipated excess cash. We hypothesize that if there is an advisory role by a friendly board, firms with such a board will allocate excess cash more towards investments aimed at increasing firm values compared to firms without a friendly board. We focus on the interaction term, which captures the effect of friendly boards on the relationship between the use of excess cash and firm value. A positive (negative) coefficient on the interaction term indicates that firms with a friendly board that used up excess cash experienced a higher (lower) firm value in the following year compared to firms without a friendly board. A positive coefficient on the interaction variable would indicate the incremental impact on value due to the advisory role of a friendly board.

Table 4 presents the regression results. In model (1) of Table 4, the estimated coefficient on lagged excess cash is negative and statistically significant at a 1% level. This suggests that a higher initial excess cash balance leads to reduced value later on for firms that deplete their excess cash throughout the year. However, we observe a complete reversal of this negative effect for firms with a friendly board. The coefficient on the interaction term is positive and statistically significant at the 1% level. This implies that the utilization of excess cash negatively impacts shareholder value only for firms without a friendly board, whereas firms with a friendly board tend to use excess cash more profitably. The result suggests that the marginal impact of utilizing one dollar of excess cash on firm value is more than 154 times greater for firms with a friendly board compared to firms without a friendly board. Model (2) controls for board and CEO characteristics variables, such as board size, CEO duality, CEO age, CEO gender, and CEO tenure. The coefficient on the interaction term remains positive and significant, confirming that firms with a friendly board utilize their excess cash more profitably compared to those without a friendly board. The results are consistent with our efficient advisory hypothesis (H3).

**Table 4.** Friendly boards and use of excess cash.

| Dependent Variable: Market-to-Book | (1) | (2) |
|---|---|---|
| Lagged excess cash | −58.025 *** | −74.329 *** |
| | (−2.81) | (−3.50) |
| Lagged excess cash x Lagged friendly board | 154.500 *** | 170.638 *** |
| | (4.21) | (4.59) |
| Lagged friendly board | −0.062 | −0.043 |
| | (−0.78) | (−0.51) |
| Lagged market-to-book | 0.545 *** | 0.537 *** |

**Table 4.** *Cont.*

| Dependent Variable: Market-to-Book | (1) | (2) |
|---|---|---|
| | (31.16) | (28.97) |
| Assets | −0.014 *** | −0.014 *** |
| | (−3.06) | (−2.83) |
| PP&E | 1.272 *** | 1.120 *** |
| | (3.51) | (2.97) |
| Institutional ownership | −0.010 *** | −0.011 *** |
| | (−4.98) | (−4.92) |
| Insider ownership | −0.008 | −0.007 |
| | (−1.36) | (−1.08) |
| CEO wealth sensitivity | 0.000 | 0.000 ** |
| | (1.53) | (2.05) |
| Board size | | −0.025 * |
| | | (−1.62) |
| CEO duality | | 0.057 |
| | | (0.76) |
| CEO age | | −0.009 |
| | | (−1.48) |
| CEO gender | | −0.192 |
| | | (−0.92) |
| CEO tenure | | 0.006 |
| | | (0.95) |
| Year and firm fixed effects | Yes | Yes |
| N | 4521 | 4271 |
| Adjusted $R^2$ | 74.81% | 74.22% |

This table reports the results of regressions of the market value on use of excess cash and friendly boards. Variable definitions are in the Appendix. The t-statics are reported in parentheses below each estimate. The labels *, **, and *** indicate statistical significance at the 1%, 5%, and 10% levels, respectively.

### 4.4. The Impact of Friendly Boards on Capital Expenditures

Next, we explore whether the impact of capital expenditures on shareholder value is influenced by the presence of friendly boards. To explore this, we employ the same framework we used in the previous section. We run a regression of market-to-book on change in capital expenditures, lagged friendly board indicator, and an interaction between the two. Because our primary focus is on substantial increases in capital expenditures, we limit our analysis to firm-year observations where the percentage increase in capital expenditures from the previous year is at least 5%. The regression equation is specified as follows:

$$
\begin{aligned}
Market\ to\ book_{i,t} = {} & \beta_0 + \beta_1 \Delta Capital\ expenditures_{i,t} + \beta_2 Lagged\ friendly\ board_{i,t} \\
& + \beta_3 Lagged\ friendly\ board_{i,t} \times \Delta Capital\ expenditures_{i,t} \\
& + \beta_4\ Lagged\ market\ to\ book_{i,t} + \beta_5 Assets_{i,t} + \beta_6\ PP\&E_{i,t} \\
& + \beta_7 Institutional\ ownership_{i,t} + \beta_8 Insider\ ownership_{i,t} \\
& + \beta_9 CEO\ wealth\ sensitivity_{i,t} + Year\ fixed\ effects \\
& + Firm\ fixed\ effects + \varepsilon_{i,t}
\end{aligned}
\tag{4}
$$

The only difference between this model and the model in the previous section is that we replace *Lagged excess cash* with $\Delta Capital\ expenditures$. To assess the impact of CEO-friendly boards on the relationship between capital expenditures and firm values, we interact lagged excess cash and lagged changes in capital expenditures. If there is an advisory role by a friendly board, firms with such a board will make more positive net

present value investments, thereby enhancing firm values, as compared to those without a friendly board. We focus on the interaction term, capturing the effect of friendly boards on the relationship between capital expenditures and firm value. A positive (negative) coefficient on the interaction term indicates that firms with a friendly board that make large capital expenditures experienced a higher (lower) firm value in the following year compared to firms without a friendly board. A positive coefficient on the interaction variable would indicate the incremental impact on value due to the advisory role of a friendly board.

The result from the analysis of Equation (4) is presented in model (1) of Table 5. The coefficient of Δ*Capital expenditures* is significantly negative, while the coefficient of the interaction term is significantly positive. The result shows that a substantial increase in capital expenditures has a significantly negative effect on shareholder value for firms without a friendly board, while the negative effect is reversed for firms with a friendly board. The result implies that the marginal impact of an additional dollar of capital expenditures on firm value is more than 2 times greater for firms with a friendly board compared to firms without a friendly board. The result suggests that firms with friendly boards invest in more profitable projects compared to firms without friendly boards. The results remain consistent in the model (2) when incorporating board and CEO characteristics variables, including board size, CEO duality, CEO age, CEO gender, and CEO tenure.

**Table 5.** Friendly boards and capital expenditures.

| Dependent Variable: Market-to-Book | (1) | (2) |
|---|---|---|
| ΔCapital expenditures | −1.003 ** | −0.820 * |
| | (−2.20) | (−1.72) |
| ΔCapital expenditures x Lagged friendly board | 2.372 *** | 2.062 *** |
| | (2.91) | (2.48) |
| Lagged friendly board | −0.122 ** | −0.114 * |
| | (−2.03) | (−1.83) |
| Lagged market-to-book | 0.483 *** | 0.463 *** |
| | (48.84) | (44.96) |
| Assets | −0.023 *** | −0.023 *** |
| | (−7.75) | (−7.45) |
| PP&E | 1.565 *** | 1.656 *** |
| | (5.46) | (5.68) |
| Institutional ownership | −0.013 *** | −0.013 *** |
| | (−8.00) | (−7.61) |
| Insider ownership | −0.003 | −0.006 |
| | (−0.75) | (−1.28) |
| CEO wealth sensitivity | 0.000 | 0.000 |
| | (−0.59) | (−0.76) |
| Board size | | −0.019 * |
| | | (−1.63) |
| CEO duality | | −0.012 |
| | | (−0.21) |
| CEO age | | 0.001 |
| | | (0.15) |
| CEO gender | | −0.093 |

**Table 5.** *Cont.*

| Dependent Variable: Market-to-Book | (1) | (2) |
|---|---|---|
| | | (−0.58) |
| CEO tenure | | 0.007 |
| | | (1.52) |
| Year and firm fixed effects | Yes | Yes |
| N | 8452 | 8028 |
| Adjusted R$^2$ | 74.37% | 74.39% |

This table reports the results of regressions of the market value on capital expenditure and friendly boards. Variable definitions are in the Appendix A. The t-statics are reported in parentheses below each estimate. The labels *, **, and *** indicate statistical significance at the 1%, 5%, and 10% levels, respectively.

### 4.5. The Impact of Friendly Boards on Investment Efficiency

In this section, we examine the effects of friendly boards on investment efficiency. Our empirical framework is adapted from Fazzari et al. (1988) and Baker et al. (2003). Investment efficiency is measured by the sensitivity of investment to Tobin's q. We estimate the following model:

$$
\begin{aligned}
Investment_{i,t} =\ & \beta_0 + \beta_1 Q_{i,t-1} \times Friendly\ board_{i,t-1} + \beta_2 CF_{i,t-1} \times Friendly\ board_{i,t-1} \\
& + \beta_3\ Friendly\ board_{i,t-1} + \beta_4 Q_{i,t-1} + \beta_5\ CF_{i,t-1} \\
& + \beta_6 Institutional\ ownership_{i,t-1} + \beta_7 Insider\ ownership_{i,t-1} \\
& + \beta_8 CEO\ wealth\ sensitivity_{i,t-1} + Year\ fixed\ effects \\
& + Firm\ fixed\ effects + \varepsilon_{i,t}
\end{aligned}
\tag{5}
$$

The dependent variable is an investment, which is measured by yearly growth in property, plant, and equipment plus yearly growth in inventory plus R&D spending, all scaled by the lagged book value of total assets. The independent variables include the following: Q, as measured by the market value of equity minus the book value of equity plus the book value of total assets, all scaled by the book value of total assets; CF, as measured by net income before extraordinary items plus depreciation and amortization plus R&D expenses, all scaled by the lagged book value of total assets; institutional ownership; insider ownership; and CEO wealth sensitivity. To examine whether CEO-friendly boards can improve investment efficiency by leveraging the input of external directors, who offer valuable advice and contribute to the decision-making process, particularly in the selection of optimal projects, we interact with the friendly board and lagged Tobin's q. The interaction term $\beta_1$ in Equation (5) is our key independent variable of interest. A positive (negative) coefficient on the interaction term $\beta_1$ indicates that friendly boards have a positive (negative) effect on the sensitivity of investment to Q. A positive coefficient on the interaction variable would suggest that investment efficiency is a mechanism through which CEO-friendly boards enhance firm values.

The result from the analysis of Equation (5) is presented in model (1) of Table 6. The estimated coefficient on the interaction term between lagged Q and friendly board is positive and statistically significant. The result shows that the investment is more sensitive to Q in firms with a friendly board compared to those without a friendly board. This implies that friendly boards can enhance investment efficiency by allowing directors to provide advice, thereby improving management's ability to select better projects. Specifically, the result shows a 50% increase in investment efficiency in firms with a friendly board compared to firms without a friendly board, calculated as the ratio of the coefficient on the interaction term (0.002) to the coefficient on lagged Q (0.004). In Model (2), we also include board and CEO characteristics variables, such as board size, CEO duality, CEO age, CEO gender, and CEO tenure. Our inference remains unaffected. The results are consistent with our efficient advisory hypothesis (H4).

**Table 6.** Friendly boards and investment efficiency.

| Dependent Variable: Investment | (1) | (2) |
|---|---|---|
| Lagged Q x Lagged friendly board | 0.002 *** | 0.003 *** |
| | (5.00) | (7.54) |
| Lagged CF x Lagged friendly board | −0.004 | −0.007 |
| | (−0.91) | (−1.38) |
| Lagged friendly board | −0.004 *** | −0.006 *** |
| | (−3.66) | (−5.66) |
| Lagged Q | 0.004 *** | 0.004 *** |
| | (15.49) | (12.64) |
| Lagged CF | 0.008 *** | 0.019 *** |
| | (2.99) | (6.62) |
| Lagged institutional ownership | | 0.000 *** |
| | | (4.65) |
| Lagged insider ownership | | 0.000 ** |
| | | (2.43) |
| Lagged CEO wealth sensitivity | | 0.000 *** |
| | | (−4.60) |
| Year and firm fixed effects | Yes | Yes |
| N | 16,053 | 14,274 |
| Adjusted $R^2$ | 86.75% | 87.29% |

This table reports the results from the OLS of investment-q sensitivity analysis. Variable definitions are in the Appendix A. The t-statics are reported in parentheses below each estimate. The labels *, **, and *** indicate statistical significance at the 1%, 5%, and 10% levels, respectively.

## 5. Conclusions

In this paper, we empirically examine the impact of CEO-friendly boards on corporate cash holdings. CEO-friendly boards impact corporate cash holdings through two primary channels: diminished monitoring by directors and improved quality of advice. The latter is facilitated by enhanced information flow from the CEO to the directors. Our findings are consistent with the efficient advisory hypothesis. CEO-friendly boards have a negative association with the level of cash holdings and a positive association with the value of cash holdings and investment efficiency.

Our study introduces a novel perspective on the influence of CEO-friendly boards on the value and utilization of corporate cash holdings. We posit that these boards play a pivotal role by facilitating seamless information flow from the CEO to the board and enhancing the quality of directors' advice. This, in turn, introduces an additional mechanism through which the value contribution of corporate cash can be further amplified.

The empirical results of this paper also offer practical implications, providing valuable insights for corporate practitioners and policymakers regarding real-world corporate governance. They suggest that we need to move beyond the traditional agency theory when evaluating the relationship between the board of directors and the CEO. At the heart of this suggestion lies the asymmetric distribution of information crucial for capital budgeting decisions between the CEO and the directors. The greater the asymmetry of this information, the more pressing the need becomes to establish collaborative information channels between the CEO and the directors by appointing CEO-friendly directors. This is particularly true for young firms. Furthermore, maintaining stability in the positions of CEO-friendly directors who have established a collaborative information channel, could be more efficient, especially when the CEO faces high information asymmetry.

Our research suggests a promising direction for future studies. It would be valuable to empirically distinguish between situations where the quality of the directors' advice is crucial and cases where the emphasis is on directors' monitoring. By delineating these

scenarios and investigating the diverse effects of CEO-friendly boards on corporate cash holdings, we anticipate uncovering enlightening insights.

**Author Contributions:** Conceptualization, H.S. and S.Y.; methodology, H.S.; software, H.S.; validation, H.S., Q.Y. and W.M.; formal analysis, H.S.; investigation, H.S. and S.Y.; resources, H.S., Q.Y. and W.M.; data curation, H.S., Q.Y. and W.M.; writing—original draft preparation H.S. and S.Y.; writing—review and editing, H.S. and S.Y.; visualization, H.S., Q.Y. and W.M.; supervision, H.S. and S.Y.; project administration, H.S., Q.Y. and W.M. All authors have read and agreed to the published version of the manuscript.

**Funding:** Our research received no external funding.

**Data Availability Statement:** The data are available from the corresponding author upon request.

**Conflicts of Interest:** The authors declare no conflicts of interest.

## Appendix A. Definition of Variables

| Variables | Definition |
|---|---|
| Friendly board | Dummy variable equals one if at least one of the independent directors is socially connected to the CEO and zero otherwise as in Schmidt (2015) and Kang et al. (2018) <br> (Data source: BoardEx) |
| Cash holdings | Natural logarithm of the ratio of cash (CHE) to sales (SALE) |
| | → ln(CHE/SALE) <br> (Data source: Compustat) |
| Assets | Book value of total assets (AT) net of cash (CHE) in billions of dollars |
| | → AT-CHE <br> (Data source: Compustat) |
| NWC | Ratio of net working capital (current assets (ACT) − current liabilities (LCT) − cash (CHE)) to the book value of total assets (AT) net of cash (CHE) |
| | → (ACT-LCT-CHE)/(AT-CHE) <br> (Data source: Compustat) |
| Cash flow | Ratio of operating income before depreciation (OIBDP) after interest (XINT), taxes (TXT), and dividends (DVC) to the book value total assets (AT) net of cash (CHE) |
| | → (OIBDP-XINT-TXT-DVC)/(AT-CHE) <br> (Data source: Compustat) |
| Cash flow volatility | Mean of the standard deviations of cash flow (Ratio of operating income before depreciation (OIBDP) after interest (XINT), taxes (TXT), and dividends (DVC) to the book value total assets (AT) net of cash (CHE)) over the previous ten years and a minimum of five years for firms in the same industry, as defined by two-digit SIC codes as in Chen et al. (2020) <br> (Data source: Compustat) |
| Market-to-book | The book value of total assets (AT) net of cash (CHE) minus the book value of equity (CEQ) plus the market value of equity (CSHO*PRCC_F), divided by the book value of total assets (AT) net of cash (CHE) |
| | → (AT-CHE-CEQ + CSHO*PRCC_F)/(AT-CHE) <br> (Data source: Compustat) |

| Variables | Definition |
|---|---|
| Capital expenditures | Ratio of capital expenditures (CAPX) to the book value of total assets (AT) net of cash (CHE) |
| | → CAPX/(AT-CHE)<br>(Data source: Compustat) |
| Acquisitions expenditures | Ratio of acquisitions expenditures (AQC) to the book value of total assets (AT) net of cash (CHE) |
| | → AQC/(AT-CHE)<br>(Data source: Compustat) |
| Leverage | Sum of long-term debt (DLTT) and debt in current liabilities (DLC) divided by the book value of total assets (AT) net of cash (CHE) |
| | → (DLTT + DLC)/(AT-CHE)<br>(Data source: Compustat) |
| R&D | Ratio of research and development expenditures (XRD) to sales (SALE). If research and development expenditure is missing, the ratio is set equal to zero. |
| | → XRD/SALE<br>(Data source: Compustat) |
| Dividend | Indicator variable that equals one in years in which a firm pays a common dividend (DVC) and zero otherwise<br>(Data source: Compustat) |
| Institutional Ownership | The aggregate percentage ownership of shares held by institutional investors that own at least 5% of outstanding shares<br>(Data source: Thomson-Reuters 13f filing) |
| Insider Ownership | The aggregate percentage ownership of common stocks held by top five executives (officers and directors)<br>(Data source: ExecuComp) |
| CEO Wealth Sensitivity | Dollar change in CEO wealth for a 100 percentage point change in firm value, divided by annual flow compensation from (Data Source: Alex Edmans (https://alexedmans.com/data/ (accessed on 16 Jan 2023).) |
| Board size | Number of directors on the board<br>(Data source: BoardEx) |
| CEO duality | Indicator variable that equals one if the CEO is the chairman of the board and zero otherwise<br>(Data source: BoardEx) |
| CEO age | Age of the CEO<br>(Data source: ExecuComp) |
| CEO gender | Indicator variable equals one for the male CEO and zero otherwise<br>(Data source: ExecuComp) |
| CEO tenure | Number of years that the CEO has been serving in the current position<br>(Data source: ExecuComp) |

| Variables | Definition |
|---|---|
| Excess cash | Ratio of the residuals from regressing cash holdings on assets, NWC, cash flow, cash flow volatility, market-to-book, capital expenditures, acquisition expenditures, leverage, RD, dividend, year fixed effects, and firm fixed effects to sales as in Dittmar and Mahrt-Smith (2007) <br> (Data source: ExecuComp) |
| PP&E | Ratio of net property, plant, and equipment (PPENT) to net assets (AT − CHE) |
| | → PPENT/(AT-CHE) <br> (Data source: Compustat) |
| Δ Capital expenditures | Change in capital expenditures (ΔCAPX) scaled by lagged net assets $((AT\text{-}CHE)_{t-1})$ |
| | → $\Delta CAPX/(AT\text{-}CHE)_{t-1}$ <br> (Data source: Compustat) |
| Investment | Yearly growth in property, plant, and equipment $(\Delta PPENT/PPENT_{t-1})$ plus yearly growth in inventory $(\Delta INVT/INVT_{t-1})$ plus R&D spending (XRD), all scaled by the lagged book value of total assets $(AT_{t-1})$ |
| | → $(\Delta PPENT/PPENT_{t-1} + \Delta INVT/INVT_{t-1} + XRD)/AT_{t-1}$ <br> (Data source: Compustat) |
| Q | Market value of equity (CSHO*PRCC_F) minus the book value of equity (SEQ) plus the book value of total assets (AT), all scaled by the book value of total assets (AT) |
| | → ((CSHO*PRCC_F)-SEQ + AT)/AT <br> (Data source: Compustat) |
| CF | Net income before extraordinary items (IB) plus depreciation and amortization (DP) plus R&D expenses (XRD), all scaled by the lagged book value of total assets $(AT_{t-1})$ |
| | → $(IB + DP + XRD)/AT_{t-1}$ <br> (Data source: Compustat) |

## Notes

[1] Economic theory suggests the following conditions for the occurrence of moral hazard problems (Mirrlees 1999). First, information among parties is uneven. Second, actions are hidden, but outcomes are visible. Third, contracts can be outcome-based. Fourth, without commitment to unseen actions, social norms do not guide party behavior. Under these conditions, the behavior of the transaction parties, constrained by these limitations, can at best achieve outcomes that are second-best, reflecting inefficiencies inherent in the situation (Holmström 1979; Mirrlees 1999). These conditions closely mirror the situation that CEOs often find themselves in, highlighting the prevalence of moral hazard in corporate governance. Hart and Holmström (1987) propose a contract-theoretic principal–agent model based on moral hazard. Similar to Mirrlees (1999), they assume a stochastic relationship between unobservable effort and observable outcomes, with the agent choosing an unobservable effort level. In this model, in fairly general situations, the agent often receives a strictly positive rent, and the principal incurs agency costs.

[2] Managerial self-indulgent activities related to the misuse of corporate cash can include pursuing unnecessary and value-decreasing growth to boost their own benefits, acquiring unrelated businesses, granting generous stock options and other forms of compensation to CEOs, and entrenching themselves through the implementation of anti-takeover provisions (Shleifer and Vishny 1997).

[3] Another vital theory related to corporate cash holdings is the pecking order theory (Myers and Majluf 1984). This theory suggests a significant information asymmetry between a firm and outside investors, making external financing costly. Consequently, a firm is incentivized to maintain sufficient retained earnings to obviate the need for external financing when lucrative investment opportunities arise, thus encouraging cash holdings. Although this theory is pivotal, it is not pertinent to the motivation of this paper, which aims to examine the relationship between CEO-friendly boards and corporate cash holdings.

[4] The existing literature on CEO-friendly boards is extensive. Previous studies on CEO-friendly boards have reported that the presence of CEO-friendly boards improves firm value by producing more patents (Kang et al. 2018) and yielding higher bidder

announcement returns when advising needs are high (Schmidt 2015). On the other hand, the presence of CEO-friendly boards can deteriorate firm value by reducing SEO announcement returns (Bhuyan et al. 2022) and diminishing labor market efficiency (Khedmati et al. 2020).

5   The underlying principle of this hypothesis is that the CEO encounters a trade-off in disclosing information to directors: The CEO sharing more information with directors can enhance the quality of the directors' advice. However, simultaneously, increased information sharing enables directors to monitor the CEO more closely. This trade-off intensifies when directors are stringent towards the CEO. Conversely, CEO-friendly directors can mitigate this trade-off, thereby facilitating information exchange between the CEO and the directors (Adams and Ferreira 2007).

6   We thank Alex Edmans for generously sharing the data with us.

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
