# Peer review of "The Relation between CEO-Friendly Boards and the Value of Cash Holdings"

_jrfm, doi:10.3390/jrfm17030113_

Round 1

Reviewer 1 Report

Comments and Suggestions for Authors

The article delves into the realm of investment efficiency, showing a positive association between CEO-friendly boards and efficient investment practices. This suggests that the positive relationship between the CEO and the board extends beyond advice and information flow, positively impacting the operational efficiency of the firm.

Recommendations for improvement to be considered by the Author:

1) The review of literature lacks specific references to the findings or methods used in the study. Providing more concrete examples and citations would strengthen the article's credibility. A comparison or discussion with other relevant studies would provide a broader context for the research. Too poor literature review.

2) The article briefly mentions the efficient advisory hypothesis without delving into the details of the methodology employed in the research. A more comprehensive explanation of the methodology would enhance the article's academic rigor.

3) The conclusion could be strengthened by summarizing the practical implications of the findings. Readers may benefit from a more explicit discussion on how these insights can be applied in real-world corporate governance. It is worth to expand on the practical implications of the study's findings and offer suggestions for corporate leaders or policymakers based on the research outcomes.

4) Ensure consistent usage of terminology throughout the article. There are instances where terms like "board dynamics" and "CEO-friendly boards" are used interchangeably, potentially causing confusion and ambiguity.

Reviewer 2 Report

Comments and Suggestions for Authors

1. The title needs revision. Make a scientific title following the call for the paper of this journal.

2. The research abstract lacks scientific depth, and it is advisable for the authors to incorporate research methods, data collection details, and statistical outcomes.

3.        Proper citations were absent in the introduction, highlighting the need for referencing throughout the study and updating outdated sources.

4. Lines 20 -28 must be revised carefully.  Several statements are unsupported by proper references i.e., citation is required to support your many statements. Is it an appropriate way of presenting old references? The organization of the study is wrong.  The organization of the study needs refining to ensure statements are substantiated with appropriate citations.

6..       The coefficient value of 0.000 raises concerns about the presentation and analysis of results.

7.       Strengthening the theoretical foundation of the research with credible sources is crucial, as the current foundation appears lacking.

8.       It is recommended that the authors thoroughly revise the content with authentic sources for a more robust theoretical framework. The research paper lacks comprehensiveness, both in terms of theoretical organization and empirical analysis, necessitating a holistic approach.

9.-      Including policy implications and underscoring the study's significance for readers and policymakers would enhance its impact. Furthermore, introducing policy implications succinctly in the abstract is essential for a comprehensive research paper.

1- Updating references with the latest research findings is also imperative for maintaining the study's relevance and credibility.

Comments on the Quality of English Language

Minor editing of English language required

Reviewer 3 Report

Comments and Suggestions for Authors

This paper is well-written with a clearly identified contribution. The role of friendly boards in the level of cash holdings and the usage of excess cash is well explored empirically and the results are new and interesting.

One minor comment about the tables is that the variable "Assets" shows as 0.000 but highly statistically significant in all tables in which is appears. Scaling the variable may help.

On a related note, more analysis of the the economic magnitude of the results could help the reader better appreciate the relative significance of a friendly board compared to other factors that drive firm cash holding policy.

Round 2

Reviewer 2 Report

Comments and Suggestions for Authors

 The authors addressed all the comments that were raised.  However, still paper has a few shortcomings that need to be addressed before acceptance.  I strongly recommend that needs to revise Appendix A. Definition of Variables as per accounting standards with proper references.  
